# Association between Lebanese Mediterranean Diet and Frailty in Community-Dwelling Lebanese Older Adults—A Preliminary Study

**DOI:** 10.3390/nu14153084

**Published:** 2022-07-27

**Authors:** Tracy Daou, Joelle Abi Kharma, Alexandra Daccache, Maya Bassil, Farah Naja, Berna Rahi

**Affiliations:** 1Department of Natural Sciences, School of Arts and Sciences, Lebanese American University, P.O. Box 13-5053, Chouran, Beirut 1102 2801, Lebanon; tracy.daou@lau.edu (T.D.); joelle.abikharma@lau.edu.lb (J.A.K.); alexandra.daccache@lau.edu (A.D.); 2Department of Human Nutrition, College of Health Sciences, Qatar University, Doha P.O. Box 2713, Qatar; bassil.maya@qu.edu.qa; 3Department of Clinical Nutrition and Dietetics, Research Institute of Medical & Health Sciences (RIMHS), College of Health Sciences, University of Sharjah, Sharjah 27272, United Arab Emirates; fnaja@sharjah.ac.ae; 4Department of Nutrition and Food Sciences, Faculty of Agriculture and Food Sciences, American University of Beirut, P.O. Box 11-0236, Riad El Solh, Beirut 1107 2020, Lebanon; 5Department of Family and Consumer Sciences, College of Health Sciences, Sam Houston State University, Huntsville, TX 77304, USA

**Keywords:** older adults, community-dwelling, frailty, Lebanese Mediterranean diet

## Abstract

Discrepancies in the characteristics of the food components of a Mediterranean diet exist based on the country of origin. In Lebanon, a traditional Mediterranean diet emphasizes the high intakes of fruits (including dried), vegetables, burghol, and dairy products. Therefore, this cross-sectional study aimed to explore the association between adherence to the Lebanese Mediterranean diet (LMD) and frailty among older adults in Lebanon. A total of 112 community-dwelling older adults aged ≥65 years were recruited. Demographic and clinical characteristics were collected through face-to-face interviews. A 61-item food frequency questionnaire (FFQ) was used to collect dietary intake data, and adherence to LMD was calculated. Physical frailty was defined by the presence of three out of the five criterion: weight loss, weakness, exhaustion, slowness, and low activity. Binary logistic regression was used to examine the relationship between LMD adherence and frailty while adjusting for several confounders. The participants’ mean age was 73 ± 12.8 and 65% were females. Sixteen (14.3%) individuals were identified as frail. Frail individuals were significantly older (*p* = 0.001), depressed (*p* < 0.001), at risk of cognitive impairment (*p* = 0.006), and reported polypharmacy (*p* = 0.003). No significant association was found between LMD adherence and frailty in fully adjusted models (OR = 0.195; 95% CI: 0.033–1.154; *p* = 0.071 when categorical and OR = 0.856; 95% CI: 0.668–1.097; *p* = 0.218 when continuous). We also performed additional analyses with a modified frailty index where house chores were not considered as part of leisure activities of the physical activity criterion. The results showed that a higher LMD adherence was associated with a significantly decreased frailty prevalence (OR = 0.123, 95% CI: 0.022–0.676, *p* = 0.016 when categorical and OR = 0.773, 95% CI 0.608–0.983, *p* = 0.036). Larger, longitudinal studies are needed to clarify the relationship between the adherence to the Lebanese Mediterranean diet and frailty in Lebanese older adults.

## 1. Introduction

The older population is considered to be the fastest expanding portion of the world population. In Lebanon, the percentage of the population aged 65 is estimated to be 7.3% and it is projected to reach 14.0% by 2030, and 23% by 2050 [1]. Several health problems such as cardiovascular diseases, diabetes, neurodegenerative diseases, and osteoporosis have been associated with aging [2]; as such, it is important to recognize the importance of successful aging in improving overall functioning and the quality of life among older adults (OA) [3]. Frailty, a geriatric syndrome, may represent a transition phase between successful aging and disability, hence it is pivotal to identify frailty as a target for implementing preventive interventions against age-related conditions [4].

Fried et al. (2001) [5] described frailty as “a biologic syndrome of decreased reserve and resistance to stressors, resulting from cumulative declines across multiple physiologic systems, and causing vulnerability to adverse outcomes”. In Lebanon, it was estimated that 15% of Lebanese OAs are considered frail in a national sample [6]; frailty prevalence increases up to 36.4% among those living in rural communities. Furthermore, a systematic review observed that frailty prevalence in low- and middle-income countries is 12.7% [7]. Studies have shown that frail individuals are at an increased risk of disabilities [8], hospitalization [9], all-cause mortality [10], chronic diseases [11,12,13,14,15], and cognitive decline [16].

Preventing frailty is an important factor for ensuring successful aging [17], and recent evidence established that nutrition is an important factor affecting its development [18,19]. In particular, adherence to a Mediterranean diet (MeDi) has been shown to reduce the incidence of frailty among OA in several Mediterranean countries like Italy [20], Spain [21], Germany [22], France [23], and Greece [24], whereas studies in Lebanon, a country on the East Basin of the Mediterranean, are still scarce [6].

In its original definition, the traditional MeDi featured a high consumption of whole cereals, legumes, vegetables, fruit and nuts, a low to mild consumption of dairy products, a low consumption of meat and poultry, and a moderate consumption of alcohol, i.e., wine during meals, whereas the consumption of olive oil was high and that of the saturated fatty acids was low [25]. The MeDi is one of the most studied and well-known dietary patterns worldwide and its beneficial effects on cardiovascular diseases, diabetes, metabolic syndrome, cancer, cognitive function, depression, and mortality are well documented [26,27,28,29], all of which are risk factors for frailty [23]. Furthermore, the beneficial effects of the MeDi on frailty are mediated by high intakes of fruits and vegetables, rich sources of antioxidants, vitamins and minerals [30] and omega-3 fatty acids, wine, and whole grains [31,32].

Although the evidence for a protective effect of MeDi on frailty has been consistent, the results of the previously cited studies cannot be generalized to other Mediterranean countries since these indices used to assess adherence were population specific. In fact, the characteristics of this pattern are quite varied as significant differences exist in what relates to types of meat (processed meat, red meat, and poultry), amount of fish, types and amount of fat or oil, types of dessert and sugar consumption, and amount and types of alcohol consumption [33]. Furthermore, these discrepancies may exist due to economic, religious, social, and cultural factors [34]. In Lebanon, an index for assessing adherence to the Middle Eastern version of the Mediterranean diet was developed from the Traditional Lebanese dietary pattern [33]. Specifically, the Lebanese MeDi (LMD) has been shown to be distinct from other Mediterranean diets by the inclusion of dried fruits and burghol, which are considered traditional Lebanese foods, and the exclusion of red meat, fish, and alcohol. Burghol, or crushed whole wheat, is a characteristic of the traditional food heritage of Lebanon and several other Eastern Mediterranean countries. Furthermore, although dairy products are considered a detrimental food component in other indices and their consumption should be limited, they are regarded as favorable foods in the LMD [33,35]. Although the only Lebanese study that examined the association of dietary patterns and frailty used three dietary patterns based on 20 predefined categories based on similarities in nutrient composition and consumption characteristics [6], the relationship between the LMD and frailty remains to be examined. Accordingly, our objective was to explore the association between the apriori dietary index-LMD and frailty, in a sample of community-dwelling Lebanese OA.

## 2. Materials and Methods

### 2.1. Study Design and Participants

The present study has a cross-sectional study design that included participants from urban communities in Lebanon. The convenient sample consisted of men and women living in communities across Lebanon who were recruited between September 2019 and February 2020 on a voluntary basis to participate in the study. The initial starting point was randomly chosen and then recruitment continued via the snowballing technique whereby the participants who agreed to participate in the study referred the trained dietitians to neighbors or friends residing in the specified areas of data collection. Those people were contacted and were asked if they are interested in participating in our study after meeting the defined eligibility criteria. One hundred thirty-five older adults were randomly approached by trained dietitians and asked if they were willing to participate. Of those approached, 112 participants agreed to participate and met the following inclusion criteria: ≥65 years old, able to understand Arabic, and living independently at home. Participants were excluded from the study if they met any of the following criteria: reported severe neurological or psychiatric disorders, suspected cognitive impairment (score < 3 on Mini-Cog), unable to walk independently and safely or using canes, history of bilateral hip replacements, any event in the last year which had a substantial impact on dietary intake and cognitive function (including death or illness of a family member), as well as currently diagnosed cancer patients.

Written consent was provided by participants and anonymity was ensured. Participants were asked to read the informed consent carefully and questions were answered verbally by the interviewer. Participants had the right to withdraw consent and remove themselves from the study at any point. All procedures performed in this study were in accordance with the ethical standards of the Institutional Review Board at the Lebanese American University (IRB #: LAU.SAS.BR4.23/Jul/2019).

The sample size was calculated using the “A-priori Sample Size Calculator for Multiple regression” with a medium level anticipated effect size (f2) of 0.15, a statistical power level of 0.8, 8 predictors, and a probability level of 0.05. This specification resulted in a total sample size of 108. An additional 25% was added to the sample size to account for incomplete data, leaving an intended final sample size of 135 participants.

### 2.2. Dietary Assessment

Nutritional data of all participants was collected by trained dietitians through a face-to-face interview using a semi-quantitative food frequency questionnaire (FFQ). The FFQ used is derived from prior studies conducted among the Lebanese population, and the cultural sensitivity and clarity of this FFQ have been previously tested by a panel of nutritionists on a sample population [33,35,36]. The FFQ consists of 61 items for the commonly consumed foods in Lebanese households. Subjects were asked to indicate the frequency of consumption (per day, per week, per month, per year, or never) for all 61 items along with the standard portion size (1cup, 1 piece, 1 teaspoon…) over the past year. Olive oil intake was assessed as an individual item, rather than in combination with the “vegetable oil” food item as this food item is a component of the LMD.

The LMD score was calculated based on Naja et al., (2015) [33]. Briefly, the LMD consists of nine food items: fruits, vegetables, legumes, olive oil, burghol (crushed whole wheat), milk and dairy products, starchy vegetables, dried fruits, and eggs. The LMD score was calculated by measuring the number of servings consumed weekly for all nine food items. Each item was divided into tertiles, and a value of 1, 2, and 3 was assigned to the first, second, and third tertiles of consumption, respectively. The sum of points received for each food item was calculated. The range for possible scores was between 9 and 27, with a score >18 indicating high adherence to LMD while a score ≤18 indicating low adherence [33]. This index is distinct from other Mediterranean diet indices by the inclusion of dried fruits, dairy products, and burghol, which are considered traditional Lebanese foods.

### 2.3. Frailty Assessment

Frailty was defined following the Cardiovascular Health Study frailty index [5], the recommended tool by the International Conference of Frailty and Sarcopenia Research [37]. Nevertheless, minor modifications were made following previous publications [23,38]. Briefly, (1) Unintentional weight loss was described as the self-reported loss of 4.5 kg in the past year. For participants who couldn’t recall if they lost weight in the previous year, a BMI < 21kg/m^2^ was used instead as an indicator for the weight loss criteria [23]; (2) Exhaustion was evaluated using the following statements from the Arabic Center for Epidemiologic Studies Depression Scale (CES-D): “I felt that everything I did was an effort” and “I could not get going” [39]. Participants were considered frail for this criterion when they answered “a moderate amount of the time” or “most of the time” to either statement; (3) Muscle strength was assessed using a Jamar ^®^ dynamometer (Rehab Mart Homecare, Singapore, Singapore), with participants sitting upright and the arm flexed at 90 degrees. The average of three measurements for the dominant hand was recorded. Low muscle strength was considered as the lowest 20th percentile of the population, stratified by BMI quartiles and gender [5]; (4) Walking speed was determined based on the time (in seconds) taken to cover a distance of 6 m, adjusting for height and gender. Participants in the slowest 20th percentile were considered frail for this criterion [5]; (5) Physical activity was assessed via an open-ended questionnaire. Low physical activity was defined as <1 h of sports activities or <3.5 h of leisure activities per week as previously described [23,38].

Older adults having three or more criteria were classified as frail whereas those having two or less were classified as non-frail. Prefrail participants were identified when they had one or two of the five criteria.

When exploring the types of physical activities performed by the participants, we observed that physical activities performed were mainly classified as leisure activities such as leisure walking, house chores, gardening, and babysitting and were mostly driven by house chores. Based on this observation, we decided to remove the house chores from leisure activities and recalculate the physical activity criterion of frailty and calculate a modified frailty index with the adjusted physical activity criterion.

### 2.4. Sociodemographic and Lifestyle Characteristics

Trained dietitians conducted face-to-face interviews with all participants. Multi-component questionnaires for socio-demographic and lifestyle characteristics included questions on: age, sex, marital status, education level, and socioeconomic status. In addition, participants were asked about their smoking habits. Body mass index (BMI) was calculated using self-reported weight and height; a strong correlation between measured and self-reported weight and height have been previously reported [40,41]. Older adults with a BMI < 23 kg/m^2^ were classified as underweight, 24–30 kg/m^2^ as healthful weight, and >30 kg/m^2^ as overweight [42]. Participants were also asked to provide information on the presence of chronic diseases (diabetes, hypertension, cardiovascular diseases (heart failure, angina, myocardial infarction, cerebrovascular diseases), kidney disease, chronic obstructive pulmonary disease, arthritis, anemia, hyper/hypothyroidism, osteoporosis, fractures, Parkinson’s disease), polypharmacy use, and dietary supplements use and number. The Arabic version of The Rowland Universal Dementia Assessment Scale (A-RUDAS) was used to assess cognitive function [43]. Individuals with a score ≤22 were considered at risk of cognitive impairment. This cut-off point exhibited good sensitivity (83%) and specificity (85%) and the Cronbach’s *α* coefficient was 0.87 [43]. The 15-item Arabic Geriatric Depression Scale (GDS-15) was used to assess participants for depressive symptoms [44]. The 7/8 cut-off point exhibited good sensitivity (0.83) and specificity (0.91) and the Cronbach’s *α* coefficient for assessing the internal consistency reliability was found to be 0.83.

### 2.5. Statistical Analysis

Demographic and clinical characteristics were compared between frail and non-frail individuals. Differences between frail and non-frail participants were tested by Mann-Whitney or *t*-test with bootstrapping for continuous variables and Pearson-chi square tests for categorical variables.

Logistic regression models were used to estimate the odds ratio (OR) and 95% confidence intervals (95%CI) for the association between LMD score and frailty. LMD score was entered as a continuous variable and as a dichotomous variable (high vs low adherence), with low adherence taken as reference. Another set of logistic regressions was performed with the modified frailty index using the adjusted physical activity criterion.

All variables having a *p*-value < 0.2 in bivariate analysis were considered for multiple logistic regression. Accordingly, each of the following variables: age, education, marital status, RUDAS, GDS, total number of chronic diseases, and polypharmacy were entered. Nevertheless, it was observed through bivariate analysis that RUDAS was highly associated with each of age, educational level, and GDS. Based on that, we decided to omit RUDAS from the final model.

Accordingly, model 1 was adjusted for age, education, and marital status. Model 2 was additionally adjusted for polypharmacy, GDS, and total number of chronic diseases. Education was entered at 3 levels: primary or less (reference category); complementary, secondary or technical; and university or higher. There were only 2 participants who reported being “divorced/separated”, so these participants were combined with the “single” category. As such, marital status was entered at the following 3 levels; “married”, “single/divorced/separated”, “widowed”, with “married” being considered as the reference group. The total number of chronic diseases was calculated by adding each chronic disease reported by the participants. Polypharmacy was taken as a dichotomous variable, with ≥5 considered as the cutoff [45]. GDS was also taken at 2 levels; with those scoring ≤7 considered for possible depression [44].

The overall models’ goodness-of-fit was assessed using Hosmer-Lemeshow’s goodness-of-fit statistics. A *p*-value that exceeds 0.05 indicates a good model fit [46]. Two post hoc statistical power analyses for multiple regression were performed based on the data generated as follows: number of predictors = 4 and 7, R^2^ = 0.26 and 0.44, an alpha = 0.05 and N = 112 for models 1 and 2, respectively [47].

Statistical analyses were performed using the statistical package SPSS version 26.0, and significance was set at *p* < 0.05.

## 3. Results

### 3.1. Participants

Initially, 135 participants were approached to participate in this study, of which 112 met our inclusion criteria. A total of 11 participants had missing information for the FFQ. These participants were contacted again through phone calls to obtain the missing data. Ten participants had missing information on the drawing section of A-RUDAS test. We verified that the final score of these participants would remain the same since they all had a score >22 (no risk of cognitive impairment) regardless of the drawing score. Based on that, we decided to keep these participants as well. The final sample consisted of 112 OA. A total of 73 females (65.2%) and 39 males (34.8%) were included in this cross-sectional study.

### 3.2. Demographic and Clinical Characteristics

Table 1 shows the characteristics of the study participants by frailty status. In total, 96 participants (85.7%) were found to be non-frail, and 16 participants (14.3%) frail. Frail participants (median age = 71 (11.75)) were significantly older than non-frail OA (median age = 82 (11.5)) (*p* = 0.001). Although not significant, compared to non-frail participants, frail participants were more likely to be female (62.5%), less educated (62.5% primary or less), and more likely to be single/divorced/widowed (68.7%).

A statistically significant difference was observed between frail and non-frail for possible depression (*p* < 0.001), risk of cognitive impairment (*p* = 0.006), and polypharmacy use (*p* = 0.003). Other characteristics including smoking status, BMI, dietary supplements use, number of dietary supplements, number of chronic diseases, diabetes, hypertension, and cardiovascular disease are shown in Table 1.

### 3.3. Dietary Intake by Frailty Status

Table 2 highlights the differences in intake between frail and non-frail participants for each of the nine LMD components. There were no statistically significant differences between any of the food groups, but non-frail individuals had a higher intake for some favorable food items such as fruits and nuts, and vegetables. For instance, fruits and nuts intake in non-frail participants had a median intake of 13.85 (11.08) servings per week, while frail individuals had a lower intake of 9.73 (11.60) servings per week.

### 3.4. Associations between Frailty and LMD

In the model adjusted for age, education, and marital status, we did not observe any significant association between LMD adherence and frailty prevalence (Table 3). In the model additionally adjusted for polypharmacy, depression, and number of chronic diseases, the association remained not significant (model 2: OR = 0.195, 95% CI 0.033–1.154, *p* = 0.071). Furthermore, no significant difference was observed when LMD was considered as a continuous variable in both models. Nevertheless, we observed a trend towards a higher prevalence of frailty among those with lower adherence to LMD, with 68.8% of frail participants reporting low adherence. The Hosmer and Lemeshow’s goodness-of-fit test indicates that both of our models fit the data well with p-values of 0.23 and 0.88 for models 1 and 2, respectively. The observed statistical power achieved for both models 1 and 2 was 99%.

### 3.5. Association between the Modified Frailty Index and LMD

Upon checking the prevalence of individual frailty criteria, we observed that the most common criterion was physical activity, where almost all frail individuals (15 (93.8%)) were considered frail for this criterion. Based on this observation, we examined which types of physical activity contributed mostly to this criterion in our sample (data not shown). We noted that house chores were among the most reported forms of physical activities carried out and that those who are frail for the physical activity criterion do not perform any type of physical activity or house chores, whereas those classified as non-frail for this criterion mostly performed house chores. In fact, a total of 58 participants did house chores (of which 54 (94%) were females), whereby 33 (56.9%) reported doing it on a daily basis, and 16 (27.6%) reported 3–4 times per week. Since most of our sample constituted of females (65.1%), and house chores were the most frequent form of physical activity carried out by this gender, we speculated that the physical activity criterion for frailty was mainly driven by house chores. Therefore, we decided to calculate another physical activity criterion that does not consider house chores as part of leisure activities. Hence, the number of participants considered frail for physical activity, increased from 37 to 54, and the number of frail individuals increased from 16 (14.29%) to 22 (18.8%).

We performed the logistic regression with this new classification and an association between higher adherence to LMD and lower frailty prevalence was observed in model 1 (OR = 0.166, 95% CI 0.038–0.715; *p* = 0.016) and model 2 (OR = 0.123, 95% CI 0.022–0.676, *p* = 0.016) (Table 4). When considered as a continuous variable, each one unit increase in LMD score was associated with a significantly decreased prevalence of frailty in model 1 (OR = 0.826, 95% CI 0.683–0.999, *p* = 0.049) and model 2 (OR = 0.773, 95% CI 0.608–0.983, *p* = 0.036).

## 4. Discussion

In the present study, we aimed to examine the association between adherence to LMD and frailty prevalence among Lebanese community-dwelling OA. We observed that adherence to LMD was not significantly associated with frailty, assessed using the frailty phenotype as described by Fried et al. [5]. Nevertheless, a trend towards higher prevalence of frailty with lower adherence to LMD was observed. In a second set of analysis, the physical activity criterion was revised by removing house chores from leisure activities. The results with the modified frailty index showed that higher adherence to LMD was significantly associated with frailty prevalence.

To our knowledge, this is the first study to assess frailty in Lebanese OA using the LMD and the frailty index. In our study, we observed frailty and prefrailty prevalence to be at 14.3% and 58%, respectively. Our results are consistent with a systematic review assessing frailty in low and middle-income countries, in which a subgroup analysis of 30 studies showed that the pooled prevalence of frailty was 12.7% and that of prefrailty was 55.2% [7]. A recent study conducted among 352 Lebanese OAs showed similar results, with 15% of the participants found to be frail, according to the FRAIL scale [6]. Another study conducted by Boulos et al. (2016) in Lebanon measuring frailty in 1200 OA using the Study of Osteoporotic Fractures index (SOF) showed different results, and estimated frailty to be 36.4% and prefrailty 30.4% [48] which is higher than what we currently observed. Possible explanations for this difference include the difference in settings where the former study was carried out in a rural area of Lebanon, where access to healthcare may be limited, and this can render this population more vulnerable to frailty. Moreover, rural areas, being usually poorer and more affected by urban migration of young adults, might witness a higher proportion of frail individuals [6]. Lastly, the difference in indices used to assess frailty can explain this discrepancy in prevalence where it was observed that the SOF index may over screen frailty compared to the frailty index [49,50].This may be because the SOF index only requires the presence of two criteria to classify an individual as frail.

In this study, a trend towards higher prevalence of frailty in OA with lower adherence to LMD was observed, with 68.8% of frail OA reporting low adherence, although this relationship did not reach significance when frailty was assessed following the frailty phenotype developed by Fried et al. [5]. When house chores were not considered part of leisure activities, a higher LMD adherence was significantly associated with lower frailty prevalence. Nonetheless, these results should be interpreted with caution since the OR obtained is smaller than any OR reported in the literature [20,21,23,24,51,52] and could have been exaggerated. In addition, the 95% CI obtained is very wide which could indicate variance in the data. More importantly, house chores are considered as main leisure activities among the older population [53,54], and were shown to be associated with health benefits [55], hence cannot be discarded [55,56]. For the aforementioned reasons, the remaining discussion will focus mainly on the results of the analyses performed using the frailty index as developed by Fried et al. [5].

These results are in line with the observations of a recent cross-sectional analysis performed on 352 Lebanese community-dwelling OA aged ≥60 years old where the MeDi was not associated with frailty as assessed by the FRAIL scale [6]. In this study, three dietary patterns were identified: (1) a westernized-type dietary pattern (WDP), (2) a high intake/Mediterranean-type dietary pattern (HI-MEDDP), and (3) a moderate intake/Mediterranean-type dietary pattern (MOD-MEDDP). Both MEDDPs were characterized by significantly lower caloric intake, refined flour products, meat and poultry, and sugar and jams than the WDP, and a higher intake of whole breads and cereals (including burghol), fruits, vegetables, legumes, milk and dairy products, and olive, seeds and oleaginous fruits, whereas fish intake was similar in all three patterns. Therefore, both identified MEDDPs are similar to the LMD used in this study, emphasizing the intake of fruits, vegetables, burghol, and dairy products. The authors observed that compared to the MOD-MEDDP, the WDP was significantly associated with a higher risk of frailty when stratified by sex and a borderline significance when analyzing the whole sample whereas no significant difference was observed when compared with the HI-MEDDP. The authors concluded that a diet far from the traditional one appears as a key deleterious determinant of frailty. These results might indicate that in the Lebanese population a nutritional transition towards a westernized diet has more detrimental effects on frailty than the protective effects of a high MeDi or LMD adherence.

Contrary to our results, previous studies have shown a significant association between adherence to a MeDi and frailty risk [20,21,23,24]. One systematic review, including 4 studies and a total of 5789 participants with a mean follow-up of 3.9 years, observed that community-dwelling OA with higher adherence to a MeDi have a significantly decreased risk of frailty [56].

Several reasons can explain the absence of a significant association in our sample. The protective effect of the MeDi on frailty is incurred mainly through the antioxidant and anti-inflammatory effect of its food components, mainly fruits and vegetables. These two food groups are rich sources of antioxidants such as polyphenols, vitamins C and E, selenium, and carotenoids, and the deficiency of these micronutrients is associated with frailty [31]. Furthermore, it was demonstrated that community-dwelling OAs who consume 3 portions of fruits per day and 2 portions of vegetables per day have reduced frailty incidence [30,57]. In our sample, both frail and non-frail participants reported an intake lower than the recommended intake of 3 and 2 portions per day for fruits and vegetables, respectively. Furthermore, these intakes were not significantly different between frail and non-frail groups.

Another mechanism that can mediate the relationship between MeDi and frailty is its potential ability to improve physical function and increase protein synthesis, by emphasizing the intake of plant-based proteins (vegetables, nuts, legumes) [58,59]. In our sample, legumes intake was similar between frail and non-frail groups. Besides the protective effect of fruits, nuts, legumes, and vegetables on frailty, Kojima (2018) additionally attributes the positive effects of the MeDi to whole grains, wine, and fish [58]. High intake of whole grains has been in fact related to lower risk of frailty [60,61]. In the LMD, burghol is what accounts for whole grains, and its consumption was found to be very low in our sample. Furthermore, moderate alcohol intake, mainly wine, when consumed with meals, was shown to decrease frailty [62]. This effect is mainly related to resveratrol that is highly concentrated in wine, as evidenced by Rabassa et al. (2015) [63]. Moreover, a recent study found that dietary fish intake has positive effects on frailty incidence, possibly due to the role of omega-3 PUFAs, which are known to have anti-inflammatory properties [64]. In this sample of OAs, both fish and wine intakes were very low, with consumption reported to be <1 serving/week (data not shown). Based on that, we speculate that the absence of association can be attributed to the lower than the recommended intake of several food components in the LMD that have been shown to be integral in reducing the risk of frailty. Moreover, in the traditional MeDi, milk and dairy products are considered as detrimental food components, whereas in the LMD they are regarded otherwise. Nevertheless, a recent analysis of the 3C-Study-Bordeaux did not observe any significant associations between total dairy products and their sub-types (milk, fresh dairy products, and cheese) and frailty prevalence and risk [65]. These results should be interpreted with caution as their generalizability to the Lebanese population and the LMD is limited since the type of fresh dairy products and cheese consumed in Lebanon is completely different than that consumed by the French population.

Moreover, although longitudinal studies observed the benefits of long-term adherence to MeDi on frailty, we were not able to duplicate such findings in our cross-sectional design. Although the FFQ is designed to capture long-term intake to overcome seasonality changes and day-to-day variations, it might not have reflected dietary behaviors beyond its assigned one-year period. This is noteworthy to mention as the current diet of OA might not indicate a lifetime dietary pattern. In fact, dietary habits of OA can be modified to either have a better or worse diet quality. On one hand, changes due to age (lower appetite, changes in taste and smell, declining physical function), food access (food cost, support with food, maintaining independence), living alone (cooking for one, eating alone, shopping for one), and relationship with food (food variety, eating what you want, dieting) can lead to worsening the older adults’ diets [66]. On the other hand, the diet quality can be enhanced when OA change their dietary behaviors to decrease the risk or manage the diagnosis of chronic diseases [67]. We speculate these changes might have been occurring in our sample, hence the non-significant differences in the individual LMD food groups between non-frail and frail participants. Therefore, longitudinal studies reflecting long-term dietary habits are needed to better assess the role of LMD adherence on frailty among Lebanese OA.

In addition, it is possible that the null association observed can be due to the sample size. In the literature, the sample size of studies assessing the relationship between frailty and dietary patterns ranged from 192 to 9861 participants, which is a sample size larger than ours [22,52].The low number of frail participants reported in this sample could have also attenuated the relationship. In fact, when we removed house chores from leisure activities accounting for PA, the number of frail participants increased from 16 to 22. Furthermore, the association between LMD as a continuous or categorical variable and frailty became significant. Nonetheless, as previously discussed, these results should be considered with caution as they have limited generalizability.

This study has some methodological limitations that should be considered. First, the number of detected frailty cases was low in our sample, compared to non-frail cases. Therefore, we speculate that the imbalance between both groups might lead to underpowered comparisons, hindering the observation of real differences, if any. Furthermore, the cross-sectional nature of this analysis does not allow us to infer causation. Moreover, we took a convenient sample which could result in selection bias, since no randomization was applied for the recruitment of participants. This also indicates that the results of this study are not generalizable to the Lebanese OA population since lifestyle and dietary habits can differ between one region and the other [68]. Nevertheless, the participants were recruited from two major urban areas in Lebanon, especially Beirut and Mount Lebanon where a high percentage of the Lebanese reside (36.7% and 30.1%, respectively). In addition, the LMD score is based on tertile intakes of the studied population which further limits its generalizability in characterizing those who are low vs high adherent to this diet. Furthermore, measurement and recall errors, and social desirability bias cannot be ruled out as several measures such as dietary data, height, and weight were self-reported. Moreover, although the FFQ was not validated in the current population, it has been extensively used in different studies across Lebanon. Despite these limitations of the FFQ method, it remains one of the most suitable dietary assessment tools [69]. In addition, the FFQ was administered by trained dietitians rather than being self-administered. This approach provides several advantages, as FFQ self-administration requires a literate population and may result in inconsistent interpretations and lower response and completion rates, each of which may jeopardize the validity of the data [69]. This is noteworthy to mention as our sample had a low level of education with 68.7% having middle school education. Moreover, we were not able to adjust for energy intake; however, BMI was not significantly different between both frail and non-frail participants. Nevertheless, we believe it would have minimal effect as BMI is highly correlated with energy intake and one study observed that results remained unchanged when comparing models adjusted for energy intake vs those not adjusted for this variable [24]. Lastly, although we adjusted for several confounders in our models, some residual confounding factors cannot be dismissed.

Despite these limitations, the strengths of this study should be highlighted. To our knowledge, we are among the first to assess dietary patterns, and in particular, the LMD in relation to frailty among Lebanese community-dwelling OAs, within a population-based setting, while adjusting for major confounders. Furthermore, we used the Cardiovascular Health Study frailty index, which is the recommended tool by the International Conference of Frailty and Sarcopenia Research [37]. In addition, all other tools that were used in data collection were validated among the Lebanese population. In addition, dietitians conducting the interviews had completed a three-day workshop, and received training on proper data collection, which can reduce interviewing bias. Lastly, a recent systematic review aiming to characterize the adherence to the MeDi in the general adult population of Mediterranean countries observed that most identified studies related to MeDi pertained to European Mediterranean countries with fewer studies from the Middle Eastern and North African Mediterranean countries, showing a clear gap in this research area [70]. Therefore, the current study helps with the understanding of different MeDi types and allows the comparison of dietary habits between different countries around the Mediterranean basin.

## 5. Conclusions

In conclusion, the present study showed that frailty prevalence in our sample of community-dwelling Lebanese OAs is in accordance with that observed in other low-middle income countries. Nevertheless, no significant associations were observed between frailty prevalence and adherence to the Lebanese Mediterranean diet. Despite the null association, the MeDi in general is established as one of the healthiest dietary patterns and it is recommended for lower morbidity and mortality among OAs. Nevertheless, studies on the LMD are scarce and future prospective longitudinal cohorts with a large sample size are needed to clarify the relationship between LMD and frailty, such that national policies can be targeted towards preserving and supporting this traditional diet. In the meantime, OAs are recommended to follow a general Mediterranean-type diet because of its well-recognized benefits on morbidity and mortality.

## Figures and Tables

**Table 1 nutrients-14-03084-t001:** Demographic and clinical Characteristics of study participants by frailty status.

	All (*n* = 112)	Non-Frail (*n* = 96)	Frail (*n* = 16)	*p*
Sex, female, %	73 (65.2)	63 (65.6)	10 (62.5)	0.81
Age (years)	73 (12.8)	71 (11.8)	82 (11.5)	**0.001**
Education levels, %				0.12
Primary or less	46 (41.1)	36 (37.5)	10 (62.5)	
Complementary	31 (27.6)	27 (28.1)	4 (25)	
Secondary or technical	14 (12.5)	14 (14.6)	0 (0)	
University or higher	21 (18.8)	19 (19.8)	2 (12.5)	
Marital Status, %				0.08
Married	60 (53.6)	55 (57.3)	5 (31.3)	
Single	14 (12.5)	11 (11.5)	3 (18.7)	
Divorced/separated	2 (1.8)	1 (1.0)	1 (6.3)	
Widowed	36 (32.1)	29 (30.2)	7 (43.7)	
BMI (kg/m^2^)				
Mean (SD)	26.10 (4.28)	26.16 (4.01)	25.72	0.71
<23	24 (21.4)	20 (20.8)	4 (25.0)	0.60
23–30	74 (66.1)	65 (67.7)	9 (56.3)	
>30	14 (12.5)	11 (11.5)	3 (18.8)	
Smoking status, %				0.04
Never smoked	56 (50.0)	46 (47.9)	10 (62.5)	
Current smoker	30 (26.8)	28 (29.2)	2 (12.5)	
Past smoker	26 (23.3)	22 (22.9)	4 (25.0)	
Frail for PA criterion	37 (33.04)	22 (22.9)	15 (93.8)	**<0.001**
Depressive symptomatology, %	20 (17.9)	11 (11.5)	9 (56.3)	**<0.001**
Risk of cognitive impairment, %	21 (18.8)	14 (14.6)	7 (43.8)	**0.006**
Polypharmacy ≥5 drugs/day, %	29 (25.9)	20 (20.8)	9 (56.3)	**0.003**
Dietary supplements, %	74 (66.1)	66 (31.3)	8 (50.0)	0.16
Number of dietary supplements	1 (2)	1 (2)	0.5 (3)	0.99
Number of chronic diseases	2 (2)	2 (2)	2.5 (1.75)	0.13
Diabetes, %	31 (27.7)	26 (27.1)	5 (31.3)	0.77
Hypertension, %	66 (58.9)	55 (57.3)	11 (68.8)	0.42
Cardiovascular diseases, %	38 (33.9)	30 (31.3)	8 (50.0)	0.16
LMD score (9–27)	18 (4.75)	18 (4.75)	18 (4.5)	0.30
LMD categories, %				0.24
Low LMD, *n* (%) (≤18)	62 (55.4)	51 (53.1)	11 (68.8)	
High LMD, *n* (%) (>18)	50 (44.6)	45 (46.9)	5 (31.2)	

All data are presented as *n* (%) except for age, dietary supplements, number of chronic diseases, and LMD score, where median (Q3-Q1) were presented. Mean (SD) is presented for BMI as a continuous variable. Differences between frail and non-frail participants tested by Mann-Whitney or chi square tests depending on the type of the variable. Significant values *p* < 0.05. Bold values are statistically significant (*p* < 0.05).

**Table 2 nutrients-14-03084-t002:** Intake for each component of the LMD between frail and non-frail.

	All Participants (*n* = 112)	Non-Frail (*n* = 96)	Frail (*n* = 16)	*p*
**Fruits and nuts**	12.99 (11.51)	13.85 (11.08)	9.73 (11.60)	0.185
**Vegetables**	11.13 (10.69)	11.13 (10.5)	10.38 (11.75)	0.732
**Starchy Vegetables**	1.5 (2)	1.5 (2.18)	1.5 (2)	0.421
**Dried Fruits**	0 (0.25)	0 (0.25)	0 (0)	0.034
**Legumes**	1 (0.5)	1 (0.5)	0.75 (1.5)	0.420
**Milk and dairy**	18.06 (14.98)	18.06 (14.59)	16.38 (18.14)	0.589
**Eggs**	2 (2)	2 (2)	2 (2.75)	0.854
**Olive Oil**	7 (7)	7 (8)	7 (4)	0.091
**Burghol**	0.5 (0.75)	0.5 (0.75)	0.75 (0.69)	0.797

All data are presented by median (IQR-Interquartile Range). Intake is presented by serving/week. Differences between non-frail and frail is tested by student *t*-test with bootstrapping.

**Table 3 nutrients-14-03084-t003:** Association between LMD adherence and frailty prevalence among Lebanese community-dwelling older adults.

	LMD as Continuous Variable		LMD as Categorical Variable		
			Low LMD Adherence	High LMD Adherence	
	OR (95% CI)	*p*		OR (95% CI)	*p*
*n* frail/total	16/112		11/62	5/50	
Model 1	0.890 (0.731–1.083)	0.244	Ref	0.235 (0.050–1.116)	0.069
Model 2	0.856 (0.668–1.097)	0.218	Ref	0.195 (0.033–1.154)	0.071

LMD Lebanese Mediterranean diet, OR Odds ratio, CI Confidence Intervals. Model 1: Model adjusted for age, education level, and marital status. Model 2: Model 1 + additional adjustment for GDS, total number of chronic diseases, polypharmacy.

**Table 4 nutrients-14-03084-t004:** Association between LMD adherence and the modified frailty index among Lebanese community-dwelling older adults.

	LMD as Continuous Variable		LMD as Categorical Variable		
			Low LMD Adherence	High LMD Adherence	
	OR 95% CI	*p*		OR 95% CI	*p*
*n* frail/total	22/112		16/62	6/50	
Model 1	0.826 (0.683–0.999)	**0.049**	Ref	0.166 (0.038–0.715)	**0.016**
Model 2	0.773 (0.608–0.983)	**0.036**	Ref	0.123 (0.022–0.676)	**0.016**

LMD Lebanese Mediterranean diet, OR, Odds ratio; CI, Confidence Intervals. Model 1: Model adjusted for age, education level and marital status. Model 2: Model 1 + additional adjustment for GDS, total number of chronic diseases, polypharmacy.

## Data Availability

Data described in the manuscript, code book and analytic code will be made available upon request.

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
