# Peer review of "Association between Lebanese Mediterranean Diet and Frailty in Community-Dwelling Lebanese Older Adults—A Preliminary Study"

_nutrients, 2022, doi:10.3390/nu14153084_

Round 1

Reviewer 1 Report

The manuscript is very well written, logical to follow, and enjoyable to read. The investigators have been diligent in their design and collection of the data. However, I have a few minor comments.

Introduction lines 50-55: The Mediterranean diet is introduced but not fully described. There are several different versions of the Mediterranean diet, which the authors acknowledge, but, again, don’t provide a clear definition of the diet. How does it differ, for example, from the Western or Chinese diets? Everyone thinks they know what the Mediterranean diet is (for example, putting olive oil on everything), but it is never very clearly defined. A few descriptive sentences should be added.

Materials and Methods line 101 to 110: Does the diet collection method measure current or past intake (diet- history)?

Materials and Methods line 194: It is unclear why removing house chores from the activity data is called sensitivity analysis. I thought removing house chores was an interesting way to look at the activity data. I just don’t understand naming it sensitivity analysis.

Results line 238 to 239: What is the number in parathesis in the Table? Table 1 parathesis contains the percent, but that does not fit Table 2.

Discussion line317: I believe the 2) is missing before high intake/Mediterranean…

Discussion line 349: Should Medi be MeDi?

Discussion: Does the method used to collect food information reflect the current intake or long-term dietary pattern? No significant differences were found between LMD and frailty. If the dietary assessment reflects current intake, it is understandable there was no connection. However, frailty may be more related to dietary patterns over a lifetime. A few sentences on the impact of long-term dietary habits should be added.

Author Response

Reviewer 1:

The manuscript is very well written, logical to follow, and enjoyable to read. The investigators have been diligent in their design and collection of the data. However, I have a few minor comments.

We would like to thank the reviewer for the positive feedback, and we will address all the concerns below.

Introduction lines 50-55: The Mediterranean diet is introduced but not fully described. There are several different versions of the Mediterranean diet, which the authors acknowledge, but, again, don’t provide a clear definition of the diet. How does it differ, for example, from the Western or Chinese diets? Everyone thinks they know what the Mediterranean diet is (for example, putting olive oil on everything), but it is never very clearly defined. A few descriptive sentences should be added.

Thank you for pointing that out. We have added the following definition of the traditional Mediterranean diet: “In its original definition, the traditional Mediterranean Diet featured a high consumption of whole cereals, legumes, vegetables, fruit and nuts, a low to mild consumption of dairy products, a low consumption of meat and poultry and a moderate consumption of alcohol, i.e. wine, during meals. The consumption of olive oil was high while the saturated fatty acids (SFA) intake was low (Trichopoulou et al., 2006)” (Lines 64-68)

Materials and Methods line 101 to 110: Does the diet collection method measure current or past intake (diet- history)?

The FFQ usually inquires about the current eating patterns during the past year. The participants were asked to indicate their usual intake of the food items during the past year. So as an example, if they consume 3 apples daily, the trained dietitian noted daily. Furthermore, usually in epidemiology, an FFQ is designed to describe the long-term intake of participants to overcome seasonality changes and day-to-day variations, that way the response provides a full cycle of seasons hence the responses should be independent of the time of year (Willet W (2013) Nutritional epidemiology. 3rd ed. New York: Oxford University Press). It was also noted that true intake is usually the average intake over a long period of time, i.e. one or more years, rather than intake over few days or weeks since diets tend to be reasonably correlated from year to year (Willet W, 2013, Nutritional epidemiology. 3rd ed. New York: Oxford University Press). Therefore, most validated FFQs in the literature assess frequency in reference to the preceding year (Al-Shaar et al., 2021; Cade et al., 2004; Sierra-Ruelas et al., 2021). Thus, we do believe that the diet collection measure current intake.

Materials and Methods line 194: It is unclear why removing house chores from the activity data is called sensitivity analysis. I thought removing house chores was an interesting way to look at the activity data. I just don’t understand naming it sensitivity analysis.

Following this comment and since a similar concern was raised by reviewer 3, we decided to remove these analyses as sensitivity, and we added them to the main analysis (Lines 174-179; 209-210). We also interpreted them at the beginning of the discussion (Lines 351-361).

Results line 238 to 239: What is the number in parathesis in the Table? Table 1 parathesis contains the percent, but that does not fit Table 2.

It represents the interquartile range. It was corrected in the footnote of the table, and we also added the IQR for the whole sample.

Discussion line317: I believe the 2) is missing before high intake/Mediterranean…

We thank the reviewer for pointing out this mistake and we apologize for it. 2) is added before high intake/Mediterranean

Discussion line 349: Should Medi be MeDi?

We also apologize for this typo. It has been corrected now.

Discussion: Does the method used to collect food information reflect the current intake or long-term dietary pattern? No significant differences were found between LMD and frailty. If the dietary assessment reflects current intake, it is understandable there was no connection. However, frailty may be more related to dietary patterns over a lifetime. A few sentences on the impact of long-term dietary habits should be added.

We agree with the reviewer’s comment, and we have added the following to our discussion: “Moreover, while longitudinal studies observed the benefits of long-term adherence to MeDi on frailty, we were not able to duplicate such findings in our cross-sectional design. Although the FFQ is designed to capture long-term intake to overcome seasonality changes and day-to-day variations, it might not have reflected dietary behaviors beyond its assigned one-year period. This is noteworthy to mention as the current diet of older adults might not indicate a lifetime dietary pattern. In fact, dietary habits of older adults can be modified to either have a better or worse diet quality. On one hand, changes due to age (lower appetite, changes in taste and smell, declining physical function), food access (food cost, support with food, maintaining independence); living alone (cooking for one, eating alone, shopping for one); and relationship with food (food variety, eating what you want, dieting) can lead to worsening the older adults’ diets (Whitelock & Ensaff, 2018). On the other hand, the diet quality can be enhanced when older adults change their dietary behaviors to decrease the risk or manage the diagnosis of chronic diseases (Zhou et al., 2018). We speculate these changes might have been occurring in our sample, hence the non-significant differences in the individual LMD food groups between non-frail and frail participants. Therefore, longitudinal studies reflecting long-term dietary habits are needed to better assess the role of LMD adherence on frailty among Lebanese older adults” (Lines 423-439).

Al-Shaar, L., Yuan, C., Rosner, B., Dean, S. B., Ivey, K. L., Clowry, C. M., . . . Rimm, E. B. (2021). Reproducibility and Validity of a Semiquantitative Food Frequency Questionnaire in Men Assessed by Multiple Methods. Am J Epidemiol, 190(6), 1122-1132. https://doi.org/10.1093/aje/kwaa280

Cade, J. E., Burley, V. J., Warm, D. L., Thompson, R. L., & Margetts, B. M. (2004). Food-frequency questionnaires: a review of their design, validation and utilisation. Nutr Res Rev, 17(1), 5-22. https://doi.org/10.1079/nrr200370

Sierra-Ruelas, É., Bernal-Orozco, M. F., Macedo-Ojeda, G., Márquez-Sandoval, Y. F., Altamirano-Martínez, M. B., & Vizmanos, B. (2021). Validation of semiquantitative FFQ administered to adults: a systematic review. Public Health Nutr, 24(11), 3399-3418. https://doi.org/10.1017/s1368980020001834

Trichopoulou, A., Corella, D., Martinez-Gonzalez, M. A., Soriguer, F., & Ordovas, J. M. (2006). The Mediterranean Diet and Cardiovascular Epidemiology. Nutrition Reviews, 64(10), S13-S19. https://doi.org/https://doi.org/10.1111/j.1753-4887.2006.tb00258.x

Whitelock, E., & Ensaff, H. (2018). On Your Own: Older Adults' Food Choice and Dietary Habits. Nutrients, 10(4). https://doi.org/10.3390/nu10040413

Willet, W. (2013). Nutritional Epidemiology (W. Willet, Ed. 3rd ed.). Oxford University Press.

Zhou, X., Perez-Cueto, F. J. A., Santos, Q. D., Monteleone, E., Giboreau, A., Appleton, K. M., . . . Hartwell, H. (2018). A Systematic Review of Behavioural Interventions Promoting Healthy Eating among Older People. Nutrients, 10(2). https://doi.org/10.3390/nu10020128

Reviewer 2 Report

The manuscript entitled ‘Association between Lebanese Mediterranean Diet and Frailty in Community-Dwelling Lebanese Older Adults’ presents interesting issue, however some corrections are needed.

-        Taking into account the number of subjects and obtained results – authors should add ‘preliminary study’ into the title.

-        ‘Successful aging strongly emphasizes physical function and/or disability, both 40 of which are factors that appear to be strongly associated with frailty‘ – some sentences seem to be awkward.

-        Lines 37-38 ‘Several health problems 37 have been associated with aging [2]’ – please specify it

-        Lines 76-78 – ‘The convenient sample consisted of men and women living in communities across Lebanon who were recruited between September 2019 and February 2020 on voluntary basis to participate in the study’ - More detailed information about recruitment procedure should be presented.

-        Lines 80-81 ‘…and met the inclusion criteria.’ – These criteria were listed in lines 92-93. I think it would be better to describe it here.

-        Lines 96-97 – ‘history of bilateral hip replacement’ – why it was a exclusion criteria? How I t could influence the obtained results?

-        Lines 102-105 – ‘The FFQ used is derived from prior studies conducted among the Lebanese population, and the cultural sensitivity and clarity of this FFQ has been previously tested by a panel of nutritionists on a sample population [27-29].’ – please add the information if this FFQ was validated against a three-day dietary record (3DR) or 24-h dietary recall, as well as the level of under or overestimation of several nutrients.

-        Line 108 - ‘over the past year.’ – this is a quite long period – it need to be addressed in the discussion (and indicated in limitation section)

-        Line 111 – ‘The LMD score was calculated based on Naja et al., (2015) [27].’ – More information is needed about the validity and reliability of each measure. Additionally, any limitations in reliability and validity need to be addressed in the discussion.

-        Line 149-150 ‘Body mass index (BMI) was calculated using self-reported weight and height;’ – it trained dietitians conducted face-to-face interviews with all participants – why weight and height were self-reported – this is unclear and unjustified. This aspect must be explained!

-        The Arabic version of The Rowland Universal Dementia Assessment Scale (A-RUDAS), – More information is needed about the validity and reliability of each measure. Additionally, any limitations in reliability and validity need to be addressed in the discussion.

-        Some international context (e.g. in the discussion section) should be presented – instead of this this manuscript has rather national than international impact.

-        Moreover, I'm not sure what is a scientific impact of this research 

-         

-         

-         

Author Response

Reviewer 2:

The manuscript entitled ‘Association between Lebanese Mediterranean Diet and Frailty in Community-Dwelling Lebanese Older Adults’ presents interesting issue, however some corrections are needed.

We would like to thank the reviewer for the corrections in order to enhance our manuscript, and we will address all the concerns below.

-        Taking into account the number of subjects and obtained results – authors should add ‘preliminary study’ into the title.

Thank you for this comment. While we have acknowledged in our limitations that our sample size is small and might not be representative of Lebanese population in general, we believe that the present study is not a preliminary one for the following reasons:

  • The sample size is enough powered according to the sample size calculations presented in the manuscript
  • Our sample was recruited from two urban areas in Lebanon especially Beirut and Mount Lebanon where a high percentage of the Lebanese reside in these two areas (36.7% and 30.1%), therefore enhancing the representation of the Lebanese population.
  • Preliminary studies are usually conducted to evaluate the feasibility, acceptability or cost of an intervention or specific methods to be used in an intervention. They are also sometimes needed to calculate the sample size (Smith et al., Field Trials of Health Interventions, Toolbox, 2015, 3rd Ed, Oxford University Press). And we believe that none of these reasons are applicable to our study.

-        ‘Successful aging strongly emphasizes physical function and/or disability, both 40 of which are factors that appear to be strongly associated with frailty‘ – some sentences seem to be awkward.

The sentence has been changed to remove any ambiguity. We removed the previous sentence and added the following: “Frailty, a geriatric syndrome, may represent a transition phase between successful aging and disability, hence it is pivotal to identify frailty as a target for implementing preventive interventions against age-related conditions (Cesari et al., 2016).” (Lines 46-48).

-        Lines 37-38 ‘Several health problems 37 have been associated with aging [2]’ – please specify it

Few examples have been added: “such as cardiovascular diseases, diabetes, neurodegenerative diseases and osteoporosis” (line 44).

-        Lines 76-78 – ‘The convenient sample consisted of men and women living in communities across Lebanon who were recruited between September 2019 and February 2020 on voluntary basis to participate in the study’ - More detailed information about recruitment procedure should be presented.

The following information was added to better describe the recruitment procedure: “The initial starting point was randomly chosen and then recruitment continued via the snowballing technique whereby the participants who agreed to participate in the study referred the trained dietitians to neighbors or friends residing in the specified areas of data collection. Those people were contacted and were asked if they are interested in participating in our study after meeting the defined eligibility criteria.” (Lines 102-107)

-        Lines 80-81 ‘…and met the inclusion criteria.’ – These criteria were listed in lines 92-93. I think it would be better to describe it here.

We thank the author for pointing this out. We moved the inclusion/exclusion criteria to where they were mentioned first. (Lines 110-116).

-        Lines 96-97 – ‘history of bilateral hip replacement’ – why it was a exclusion criteria? How I t could influence the obtained results?

We excluded participants with history of bilateral hip replacement based on previous studies that also had this exclusion criterion (Shikany et al., 2014). Furthermore, usually when assessing frailty, inclusion criteria include the ability to be able to walk without aid. This might have underestimated the prevalence of frailty in our sample. Nevertheless, since the observed prevalence in our study is in line with that of frailty in LMICs and other studies done in Lebanon, we feel that this might had a minimal effect on the recruitment process and on our results.

-        Lines 102-105 – ‘The FFQ used is derived from prior studies conducted among the Lebanese population, and the cultural sensitivity and clarity of this FFQ has been previously tested by a panel of nutritionists on a sample population [27-29].’ – please add the information if this FFQ was validated against a three-day dietary record (3DR) or 24-h dietary recall, as well as the level of under or overestimation of several nutrients.

The FFQ used in this study was not validated against other dietary assessment methods. That said, several approaches were undertaken to improve the validity of the dietary intake data resulting from this FFQ. For instance, the food list used in the FFQ was examined by a panel of experts including a dietitian, a public health nutritionist and a nutrition epidemiologist to ensure that all foods included are commonly consumed by the target population. Furthermore, in order to improve the accuracy of the portion size estimation, a common challenge in dietary intake assessment, subjects were given the choice to report their intakes in terms of grams or as a function of a reference portion size. The reference portion, representing one standard serving, was expressed in household measures such as cups, spoons and plates. Real size photos were used to assist subjects in the identification of the reference portion size. In addition, the fact that the food frequency questionnaire was administered by a trained dietitian and not self-completed offers many advantages in that it does not require a literate population and results in consistent interpretations and higher response and completion rates, each of which may enhance the validity of the data (Willet W, 2013; Nutritional epidemiology. 3rd ed. New York: Oxford University Press and Hu et al., 1999).

-        Line 108 - ‘over the past year.’ – this is a quite long period – it need to be addressed in the discussion (and indicated in limitation section)

We understand that this might look as a long period but usually in epidemiology, an FFQ is designed to describe the long-term intake of participants to overcome seasonality changes and day-to-day variations, that way the response provides a full cycle of seasons hence the responses should be independent of the time of year (Willet W, 2013; Nutritional epidemiology. 3rd ed. New York: Oxford University Press). It was also noted that true intake is usually the average intake over a long period of time, i.e. one or more years, rather than intake over few days or weeks since diets tend to be reasonably correlated from year to year (Willet W, 2013; Nutritional epidemiology. 3rd ed. New York: Oxford University Press). Furthermore, most validated FFQs in the literature usually assess intake over a one-year period (Al-Shaar et al., 2021; Cade et al., 2004; Sierra-Ruelas et al., 2021). Therefore, we do not believe that “over the past year" is a long period, rather it is a normal period through which habitual intake can be quantified.

-        Line 111 – ‘The LMD score was calculated based on Naja et al., (2015) [27].’ – More information is needed about the validity and reliability of each measure. Additionally, any limitations in reliability and validity need to be addressed in the discussion.

The original MeDi score was developed by Trichopoulou et al. (Trichopoulou et al., 2003). They calculated the MeDi score by assigning a value of 0 or 1 to each of nine indicted components with the use of sex-specific median as cut-off. When weekly number of servings of the beneficial components was below the sex-specific median, a score of 0 was assigned and a score of 1 assigned when consumption was higher than the median. For components presumed to be detrimental, a score of 1 was assigned when weekly consumption was below the median and a score of 0 when consumption was at or above the median. Then, the MeDi score was calculated by adding the scores for each food category for each participant, ranging from 0 to 9, with higher scores indicating greater adherence. Subsequent variations of the MeDi followed the same concept, i.e summation of the scores of individual food groups identified based on statistical cut-off points (D'Alessandro & De Pergola, 2018; Feart & Barberger-Gateau, 2015).

The development of LMD followed that of the original MeDi. The identification of the food groups to be included in the LMD was based on the results of earlier investigations led by Naja et al., aiming to characterize the main dietary patterns In Lebanon (Naja et al., 2015). These investigations showed that the Lebanese traditional pattern has consistently emerged as a common pattern using factor analysis.

Out of 30 food groups/food items entered in the factor analysis, nine repeatedly loaded high on this pattern, including fruits, vegetables, legumes, olive oil, burghol (crushed whole wheat), milk and dairy products, starchy vegetables (including potato, corn and peas), dried fruits and eggs. Similar to Trichopoulou et al., method, the LMD was calculated based on the number of portions of these nine foods/food groups consumed daily. Specifically, the weekly consumption (number of servings) of each of these nine foods/ food groups were divided into tertiles and a value of 1, 2, and 3 was assigned to the first, second and third tertiles of consumption, respectively. The Lebanese pattern score was then calculated, for each subject, as the sum of points received on the consumption of the nine foods/food groups.

When compared to other European MeDi scores, the correlation between the Italian MD and the LMD reached a “good” level of association and was greater than 0.5 (r = 0.56). The LMD score had the highest correlation with scores of the Italian MD, followed by the Spanish (r = 0.30) and was least correlated with that of the French MD (r = 0.21).

Furthermore, the highest agreement was observed between the LMD and the Italian MD with 53.17 and 92.38 % of subjects classified in the same and the same or adjacent tertiles, respectively. The Kappa statistics reached 0.49 indicating a moderate agreement.

The study showed that the score of the LMD index was significantly correlated with all the selected European MeDi indices, while being the closest to the Italian MD. In addition to fruits, vegetables and olive oil, which were common denominators to indexes reported from Greece, Italy, Spain, France, the LMD index was distinctively characterized by the inclusion of burghol, dried fruits, and dairy products which are part of the traditional food heritage of Lebanon as well as other countries of the Eastern Mediterranean basin.

The main limitation of the MeDi score in general is that its computations is based on statistically identified cut-off points thus making it dependent on the sample, hence limiting its generalizability. This also applies to the LMD. The following limitation was added to the discussion: “this score is data driven since it is computed based on the tertile consumption of the current sample thus limiting the generalizability in characterizing those who are low vs high adherent to this diet.”

-        Line 149-150 ‘Body mass index (BMI) was calculated using self-reported weight and height;’ – it trained dietitians conducted face-to-face interviews with all participants – why weight and height were self-reported – this is unclear and unjustified. This aspect must be explained!

Self-reported weight and height were used instead of measured because in our culture where Islam is the religion, it is unacceptable to ask participants to be in light clothes and bare foot for weight measurements as women must be veiled in front of strangers. To be respectful for the different cultural backgrounds of our participants, we decided to use the self-reported measures especially that they are highly correlated as reported in the methodology. 

-        The Arabic version of The Rowland Universal Dementia Assessment Scale (A-RUDAS), – More information is needed about the validity and reliability of each measure. Additionally, any limitations in reliability and validity need to be addressed in the discussion.

The specificity and sensitivity for the Arabic versions of RUDAS and GDS were added: “Individuals with a score ≤ 22 were considered at risk of cognitive impairment. This cut-off point exhibited good sensitivity (83%) and specificity (85%) and the Cronbach’s α coefficient was 0.87 (Chaaya et al., 2016). The 15-item Arabic Geriatric Depression Scale (GDS-15) was used to assess participants for depressive symptoms (Chaaya et al., 2008). The 7/8 cut-off point exhibited good sensitivity (0.83) and specificity (0.91) and the Cronbach’s α coefficient for assessing the internal consistency reliability was found to be 0.83.” (Lines 196-201)

-        Some international context (e.g. in the discussion section) should be presented – instead of this this manuscript has rather national than international impact.

Following this comment, we have added the following sentences to the strengths of our study: “Lastly, a recent systematic review aiming to characterize the adherence to the MeDi in the general adult population of Mediterranean countries observed that most identified studies related to MeDi pertained to European Mediterranean countries with fewer studies from the Middle Eastern and North African Mediterranean countries, showing a clear gap in this research area (Obeid et al., 2022). Therefore, the current study helps with the understanding of different MeDi types and allows the comparison of dietary habits between different countries around the Mediterranean basin.” (Lines 478-485).

-        Moreover, I'm not sure what is a scientific impact of this research 

We would like to thank the reviewer for this feedback. The scientific impact of our research is detailed as follow: First, Lebanon is a Mediterranean country and is believed to follow a Mediterranean diet (MeDi). Nevertheless, studies have shown this diet has discrepancies based on the geographic location of the Mediterranean country in addition to economic, cultural and religious factors, hence different MeDi scores were developed. Furthermore, a recent systematic review observed that most studies related to MeDi pertained to the European Mediterranean countries, with fewer studies from the Middle Eastern and North African Mediterranean countries with a clear gap in this research area (Obeid et al., 2022). Therefore, we decided to describe the LMD and evaluate one of its beneficial effects, which is frailty. To our knowledge, no previous study has evaluated the association between LMD and frailty in Lebanon. This adds to the literature some understanding about different types of MeDi in general and the health benefits of LMD in particular among this Mediterranean population. Moreover, the older adult population in Lebanon is increasing exponentially and it has been estimated that by 2050 the percentage of adults aged 65 and above will reach 23%. Therefore, determining the dietary habits of this population and trying to improve their health through diet is pivotal. In addition, frailty is a prevalent geriatric syndrome and is higher in LMICs than developed countries. Moreover, frailty is associated with a higher risk of chronic diseases, hospitalization, and mortality. Therefore, it is important to determine approaches to prevent frailty among community dwelling Lebanese Older Adults in order to enhance the quality of their life during aging and to mitigate their risk of chronic diseases. Lastly, studies involving the beneficial effects of nutrition among Lebanese older adults are scarce in general as geriatric nutrition is a fairly new field in Lebanon. Hence, our study will be adding more information about this growing population in general, and particularly the role of nutrition as an accessible and preventive measure to enhance the aging process. 

Al-Shaar, L., Yuan, C., Rosner, B., Dean, S. B., Ivey, K. L., Clowry, C. M., . . . Rimm, E. B. (2021). Reproducibility and Validity of a Semiquantitative Food Frequency Questionnaire in Men Assessed by Multiple Methods. Am J Epidemiol, 190(6), 1122-1132. https://doi.org/10.1093/aje/kwaa280

Cade, J. E., Burley, V. J., Warm, D. L., Thompson, R. L., & Margetts, B. M. (2004). Food-frequency questionnaires: a review of their design, validation and utilisation. Nutr Res Rev, 17(1), 5-22. https://doi.org/10.1079/nrr200370

Cesari, M., Prince, M., Thiyagarajan, J. A., De Carvalho, I. A., Bernabei, R., Chan, P., . . . Vellas, B. (2016). Frailty: An Emerging Public Health Priority. J Am Med Dir Assoc, 17(3), 188-192. https://doi.org/10.1016/j.jamda.2015.12.016

Chaaya, M., Phung, T. K., El Asmar, K., Atweh, S., Ghusn, H., Khoury, R. M., . . . Waldemar, G. (2016). Validation of the Arabic Rowland Universal Dementia Assessment Scale (A-RUDAS) in elderly with mild and moderate dementia. Aging Ment Health, 20(8), 880-887. https://doi.org/10.1080/13607863.2015.1043620

Chaaya, M., Sibai, A. M., Roueiheb, Z. E., Chemaitelly, H., Chahine, L. M., Al-Amin, H., & Mahfoud, Z. (2008). Validation of the Arabic version of the short Geriatric Depression Scale (GDS-15). Int Psychogeriatr, 20(3), 571-581. https://doi.org/10.1017/s1041610208006741

D'Alessandro, A., & De Pergola, G. (2018). The Mediterranean Diet: its definition and evaluation of a priori dietary indexes in primary cardiovascular prevention. Int J Food Sci Nutr, 69(6), 647-659. https://doi.org/10.1080/09637486.2017.1417978

Feart, C., & Barberger-Gateau, P. (2015). Mediterranean Diet and Cognitive Health (Chapter 25). In C. Martin & V. Preedy (Eds.), Diet and Nutrition in Dementia and Cognitive Decline. Academic Press. https://doi.org/https://doi.org/10.1016/C2012-0-02833-5

Hu, F. B., Rimm, E., Smith-Warner, S. A., Feskanich, D., Stampfer, M. J., Ascherio, A., . . . Willett, W. C. (1999). Reproducibility and validity of dietary patterns assessed with a food-frequency questionnaire. Am J Clin Nutr, 69(2), 243-249. https://doi.org/10.1093/ajcn/69.2.243

Naja, F., Hwalla, N., Itani, L., Baalbaki, S., Sibai, A., & Nasreddine, L. (2015). A novel Mediterranean diet index from Lebanon: comparison with Europe. Eur J Nutr, 54(8), 1229-1243. https://doi.org/10.1007/s00394-014-0801-1

Obeid, C. A., Gubbels, J. S., Jaalouk, D., Kremers, S. P. J., & Oenema, A. (2022). Adherence to the Mediterranean diet among adults in Mediterranean countries: a systematic literature review. Eur J Nutr, 1-18. https://doi.org/10.1007/s00394-022-02885-0

Shikany, J. M., Barrett-Connor, E., Ensrud, K. E., Cawthon, P. M., Lewis, C. E., Dam, T. T., . . . Redden, D. T. (2014). Macronutrients, diet quality, and frailty in older men. J Gerontol A Biol Sci Med Sci, 69(6), 695-701. https://doi.org/10.1093/gerona/glt196

Sierra-Ruelas, É., Bernal-Orozco, M. F., Macedo-Ojeda, G., Márquez-Sandoval, Y. F., Altamirano-Martínez, M. B., & Vizmanos, B. (2021). Validation of semiquantitative FFQ administered to adults: a systematic review. Public Health Nutr, 24(11), 3399-3418. https://doi.org/10.1017/s1368980020001834

Smith, P., Morrow, R., & Ross, D. (2015). Preliminary Studies and Pilot Testing (Chapter 13). In Field Trials of Health Interventions (3rd ed.). Oxford University Press.

Trichopoulou, A., Costacou, T., Bamia, C., & Trichopoulos, D. (2003). Adherence to a Mediterranean diet and survival in a Greek population. N Engl J Med, 348(26), 2599-2608. https://doi.org/10.1056/NEJMoa025039

Willet, W. (2013). Nutritional Epidemiology (W. Willet, Ed. 3rd ed.). Oxford University Press. 

Reviewer 3 Report

The article deals with the relationship between the Lebanese Mediterranean diet and frailty. The study is interesting. However, I wonder if the study is sufficiently novel and solid to be published in a high impact journal like Nutrients. Anyway, I have made some suggestions and hope you find them helpful:

L30 - Something is wrong with this p-value. 

L63 - Could you please define what bulgur is (this is the first time this food item appears in the text - except for the abstract).

The introduction is short. Why is the MEdi supposed to be protective? What nutrients are in this diet that might prevent frailty? Do you have previous studies on the topic? This is an expansive diet for this population? More information on the nutritional aspect of the diet is needed.

Methods - 

L111 - Although you used a referenced method (Naja et al. 2015), I do not completely agree. I think you should do a principal component analysis to look at patterns. Is this pattern present in this population? Just counting the foods seems very arbitrary to me. What is more important consume some of these food items in great amount or have a consistent pattern, even with low consumption (in volume)? 

L203 - I know this is very hard to control for, but don't you think your sample is underpowered? The prevalence of frailty was very low (and lower than 36.4% as described in the introduction), which makes it difficult to detect significant associations. You probably have the power to run the logistic regression, but not to distinguish between people with and without frailty. 

L239 - What is IQ? The table must stand for itself.

Why did you use Mann-Whitney's U? I think it would be better to use a t-student test with bootstrapping procedure to avoid normality problems. 

L281 - I think this result is more precise than the one in Table 3. Perhaps you could also include this result in the abstract.

I found the discussion/conclusion very odd. The discussion/conclusion is mainly based on Table 3, so what is the significance of the result from Table 4? This is confusing to me.

Nutrients might be more interesting in some biological mechanisms, this should also be considered.

Author Response

Reviewer 3:

The article deals with the relationship between the Lebanese Mediterranean diet and frailty. The study is interesting. However, I wonder if the study is sufficiently novel and solid to be published in a high impact journal like Nutrients. Anyway, I have made some suggestions and hope you find them helpful:

We would like to thank the reviewer for the feedback; however, we believe that our study is novel for several reasons. First, Lebanon is a Mediterranean country and is believed to follow a Mediterranean diet (MeDi). Nevertheless, studies have shown this diet has discrepancies based on the geographic location of the Mediterranean country in addition to economic, cultural and religious factors, hence different MeDi scores were developed. Furthermore, a recent systematic review observed that most studies related to MeDi pertained to the European Mediterranean countries, with fewer studies from the Middle Eastern and North African Mediterranean countries with a clear gap in this research area (Obeid et al., 2022). Therefore, we decided to describe the LMD and evaluate one of its beneficial effects, which is frailty. To our knowledge, no previous study has evaluated the association between LMD and frailty in Lebanon. This adds to the literature some understanding about different types of MeDi in general and the health benefits of LMD in particular among this Mediterranean population. Moreover, the older adult population in Lebanon is increasing exponentially and it has been estimated that by 2050 the percentage of adults aged 65 and above will reach 23%. Therefore, determining the dietary habits of this population and trying to improve their health through diet is pivotal. In addition, frailty is a prevalent geriatric syndrome and is higher in LMICs than developed countries. Moreover, frailty is associated with a higher risk of chronic diseases, hospitalization, and mortality. Therefore, it is important to determine approaches to prevent frailty among community dwelling Lebanese Older Adults in order to enhance the quality of their life during aging and to mitigate their risk of chronic diseases. Lastly, studies involving the beneficial effects of nutrition among Lebanese older adults are scarce in general as geriatric nutrition is a fairly new field in Lebanon. Hence, our study is novel in assessing the benefits of LMD among a Mediterranean population that has not been studied yet. Therefore, we will be adding more information about this growing population in general, and particularly the role of nutrition as an accessible and preventive measure to enhance the aging process.

All other comments will be addressed below.

L30 - Something is wrong with this p-value. 

We thank the reviewer for pointing out this typo. The p-value is p=0.218 and it has been corrected.

L63 - Could you please define what bulgur is (this is the first time this food item appears in the text - except for the abstract).

We have added the following sentence to define what Bulgur is: “Burghol, or crushed whole wheat, is a characteristic of the traditional food heritage of Lebanon and several other Eastern Mediterranean countries.” (Lines 87-88). We also changed “bulgur” to “Burghol” to better reflect the Arabic name.

The introduction is short. Why is the MEdi supposed to be protective? What nutrients are in this diet that might prevent frailty? Do you have previous studies on the topic? This is an expansive diet for this population? More information on the nutritional aspect of the diet is needed.

Thank you for this comment. We have added the following paragraph to introduce the MeDi, its beneficial effects and the mechanisms.

“In its original definition, the traditional MeDi featured a high consumption of whole cereals, legumes, vegetables, fruit and nuts, a low to mild consumption of dairy products, a low consumption of meat and poultry and a moderate consumption of alcohol, i.e. wine, during meals while the consumption of olive oil was high and that of the saturated fatty acids was low (Trichopoulou et al., 2006). The MeDi is one of the most studied and well-known dietary patterns worldwide and its beneficial effects on cardiovascular diseases, diabetes, metabolic syndrome, cancer, cognitive function, depression, and mortality are well documented (Franquesa et al., 2019; Galbete et al., 2018; Guasch-Ferré & Willett, 2021; Ventriglio et al., 2020), all of which are risk factors for frailty (Rahi et al., 2018). Furthermore, the beneficial effects of the MeDi on frailty are mediated by high intakes of fruits and vegetables, rich sources of antioxidants, vitamins and minerals (García-Esquinas et al., 2016) and omega-3 fatty acids (Feart, 2019).” (Lines 67-74).

More details about the mechanisms of MeDi on frailty are extensively presented in the discussion.

Lastly, the few dietary studies that have been performed on the Lebanese older adults have been shown that this diet is consistent. In fact, several investigations done by the group of Naja et al. aiming to characterize the main dietary patterns in Lebanon using factor analysis consistently showed the emergence of a common pattern which is the Traditional LMD. Out of 30 food groups/food items entered in the factor analysis, nine repeatedly loaded high on this pattern, including fruits, vegetables, legumes, olive oil, burghol (crushed whole wheat), milk and dairy products, starchy vegetables (including potato, corn and peas), dried fruits and eggs (Naja et al., 2015). In two of those studies, the participants age range was 20 to 55 years old while in the third one the average age was 56.18 ±10.93 years. Another analysis on a sample aged 55 and above with a mean age of 66.4 ± 7.89 also observed this common pattern (Jomaa et al., 2016). In addition, a more recent study conducted by Yaghi et al., where dietary patterns were determined through cluster analysis, observed a Mediterranean diet pattern that is similar to ours, i.e. it emphasized high intakes of whole breads and cereals (including burghol), fruits, vegetables, legumes, milk and dairy products and olive, seeds and oleaginous fruits (Yaghi et al., 2021). Therefore, this pattern is consistently present in the Lebanese population. In addition, several other studies assessed adherence to MeDi among different age groups of the Lebanese population and mostly observed a moderate adherence to this diet (El Hajj & Julien, 2021; Hayek et al., 2021; Karam et al., 2021; Karam et al., 2022; Mitri et al., 2021; Naja et al., 2021).

Methods - 

L111 - Although you used a referenced method (Naja et al. 2015), I do not completely agree. I think you should do a principal component analysis to look at patterns. Is this pattern present in this population? Just counting the foods seems very arbitrary to me. What is more important consume some of these food items in great amount or have a consistent pattern, even with low consumption (in volume)? 

It is important to acknowledge that dietary studies are scarce among the Lebanese older adults. Nevertheless, the available data assessing dietary patterns among this population have shown consistency in adherence to the LMD pattern. In fact, several investigations done by the group of Naja et al. aiming to characterize the main dietary patterns in Lebanon using factor analysis consistently showed the emergence of a common pattern which is the Traditional LMD. Out of 30 food groups/food items entered in the factor analysis, nine repeatedly loaded high on this pattern, including fruits, vegetables, legumes, olive oil, burghol (crushed whole wheat), milk and dairy products, starchy vegetables (including potato, corn and peas), dried fruits and eggs (Naja et al., 2015). In two of those studies, the participants age range was 20 to 55 years old while in the third one the average age was 56.18 ±10.93 years. Another analysis on a sample aged 55 and above with a mean age of 66.4 ± 7.89 also observed this common pattern (Jomaa et al., 2016). In addition, a more recent study conducted by Yaghi et al., where dietary patterns were determined through cluster analysis, observed a Mediterranean diet pattern that is similar to ours, i.e. it emphasized high intakes of whole breads and cereals (including burghol), fruits, vegetables, legumes, milk and dairy products and olive, seeds and oleaginous fruits (Yaghi et al., 2021). Therefore, this pattern is consistently present in the Lebanese population.

Based on the reviewer’s suggestion, we explored the option of performing a PCA. A correlation matrix of 26 food groups identified according to similarities in ingredients, nutrient profile and/or culinary usage was assessed to ensure significance. Significance was taken at p<0.0.5 for the chi-square for Bartlett test of sphericity, and at a score greater than 0.6 for the Kaiser-Meyer-Olkin test (KMO). Factors were retained for an eigenvalue >1, and the inflection point of the scree plot was interpreted and Varimax rotation was chosen to rotate the factors. The factors considered were those with an absolute value of ≥0.3.

The results of the PCA analysis should be interpreted with caution for several reasons. First, our sample size does not follow the rule of thumb for PCA analysis. We entered 26 variables into the PCA, while our total observation was 112, which does not comply with the general rule of entering 1 variable for every 10 observations. Moreover, based on the 26 items entered, we observed 10 patterns that explained approximately 69% of the total variance. The number of patterns is big considering the small sample size we have. Furthermore, after examining the scree plot, a rupture was observed after the second pattern with these two patterns explaining only 30% of the total variance. Therefore, these results should be interpreted cautiously and this is why we prefer not to report the PCA analyses.

Lastly, since the LMD has been shown to a consistent dietary pattern among the Lebanese population, we feel that it is justifiable to adopt it to our population. Furthermore, the LMD considers the servings of the food groups and not just their consumption (as yes vs no). The LMD score is calculated by measuring the number of servings consumed weekly for all nine food items, and then the consumption of each item is divided into tertiles, and a value of 1, 2, and 3 was assigned to the first, second and third tertiles of consumption, respectively. This method is similar to the original MeDi developed by Trichopoulo where the MeDi score was generated as follows: a value of 0 or 1 was assigned to each food group using sex-specific medians of the population as cut-offs (Trichopoulou et al., 2003).

L203 - I know this is very hard to control for, but don't you think your sample is underpowered? The prevalence of frailty was very low (and lower than 36.4% as described in the introduction), which makes it difficult to detect significant associations. You probably have the power to run the logistic regression, but not to distinguish between people with and without frailty. 

Thank you for this comment. We have adjusted the introduction to adjust newer numbers from a recent study in a national sample. Furthermore, the frailty prevalence of 36.4% was observed in rural areas while our study is performed in performed urban areas. These differences were addressed in the discussion part. In addition, we have stated that the sample size is small, and our study might be underpowered in the limitations. 

L239 - What is IQ? The table must stand for itself. Why did you use Mann-Whitney's U? I think it would be better to use a t-student test with bootstrapping procedure to avoid normality problems. 

IQ means Interquartile range and it has been corrected in the foot note. We have also added the IQR for the whole sample. Furthermore, based on the reviewer’s suggestion, we also performed a t-student test with bootstrapping and the results did not change. Bootstrapping was done at n samples =1000. We reported the p-values obtained with bootstrapping in the table.

L281 - I think this result is more precise than the one in Table 3. Perhaps you could also include this result in the abstract.

Based on this suggestion, we added these results in the abstract: “We also performed additional analysis with a modified frailty index where house chores were not considered as part of leisure activities of the physical activity criterion. The results showed that a higher LMD adherence was associated with a significantly decreased frailty prevalence (OR=0.123, 95% CI 0.022 – 0.676, p= 0.016 when categorical and OR=0.773, 95% CI 0.608 – 0.983, p=0.036).” (Lines 30-34)

I found the discussion/conclusion very odd. The discussion/conclusion is mainly based on Table 3, so what is the significance of the result from Table 4? This is confusing to me.

Thank for this comment that allowed us to enhance our discussion. We changed to discussion in a way to reflect the analysis of table 4 and we move this part to the beginning of the discussion, we discussed these results and stated that caution should be considered when interpreting these results (Lines 351-361). After that, the discussion focused on the results of table 3.

Nutrients might be more interesting in some biological mechanisms, this should also be considered.

We agree with this comment, but the objective of our study was epidemiological. We mentioned the mechanisms by which the MeDi might have its beneficial effects on frailty in the introduction and in the discussion.

El Hajj, J. S., & Julien, S. G. (2021). Factors Associated with Adherence to the Mediterranean Diet and Dietary Habits among University Students in Lebanon. J Nutr Metab, 2021, 6688462. https://doi.org/10.1155/2021/6688462

Feart, C. (2019). Nutrition and frailty: Current knowledge. Prog Neuropsychopharmacol Biol Psychiatry, 95, 109703. https://doi.org/10.1016/j.pnpbp.2019.109703

Franquesa, M., Pujol-Busquets, G., García-Fernández, E., Rico, L., Shamirian-Pulido, L., Aguilar-Martínez, A., . . . Bach-Faig, A. (2019). Mediterranean Diet and Cardiodiabesity: A Systematic Review through Evidence-Based Answers to Key Clinical Questions. Nutrients, 11(3). https://doi.org/10.3390/nu11030655

Galbete, C., Schwingshackl, L., Schwedhelm, C., Boeing, H., & Schulze, M. B. (2018). Evaluating Mediterranean diet and risk of chronic disease in cohort studies: an umbrella review of meta-analyses. Eur J Epidemiol, 33(10), 909-931. https://doi.org/10.1007/s10654-018-0427-3

García-Esquinas, E., Rahi, B., Peres, K., Colpo, M., Dartigues, J. F., Bandinelli, S., . . . Rodríguez-Artalejo, F. (2016). Consumption of fruit and vegetables and risk of frailty: a dose-response analysis of 3 prospective cohorts of community-dwelling older adults. Am J Clin Nutr, 104(1), 132-142. https://doi.org/10.3945/ajcn.115.125781

Guasch-Ferré, M., & Willett, W. C. (2021). The Mediterranean diet and health: a comprehensive overview. J Intern Med, 290(3), 549-566. https://doi.org/10.1111/joim.13333

Hayek, J., de Vries, H., Tueni, M., Lahoud, N., Winkens, B., & Schneider, F. (2021). Increased Adherence to the Mediterranean Diet and Higher Efficacy Beliefs Are Associated with Better Academic Achievement: A Longitudinal Study of High School Adolescents in Lebanon. Int J Environ Res Public Health, 18(13). https://doi.org/10.3390/ijerph18136928

Jomaa, L., Hwalla, N., Itani, L., Chamieh, M. C., Mehio-Sibai, A., & Naja, F. (2016). A Lebanese dietary pattern promotes better diet quality among older adults: findings from a national cross-sectional study. BMC Geriatr, 16, 85. https://doi.org/10.1186/s12877-016-0258-6

Karam, J., Bibiloni, M. D. M., Serhan, M., & Tur, J. A. (2021). Adherence to Mediterranean Diet among Lebanese University Students. Nutrients, 13(4). https://doi.org/10.3390/nu13041264

Karam, J., Serhan, C., Swaidan, E., & Serhan, M. (2022). Comparative Study Regarding the Adherence to the Mediterranean Diet Among Older Adults Living in Lebanon and Syria. Front Nutr, 9, 893963. https://doi.org/10.3389/fnut.2022.893963

Mitri, R. N., Boulos, C., & Ziade, F. (2021). Mediterranean diet adherence amongst adolescents in North Lebanon: the role of skipping meals, meals with the family, physical activity and physical well-being. Br J Nutr, 1-8. https://doi.org/10.1017/s0007114521002269

Naja, F., Hwalla, N., Hachem, F., Abbas, N., Chokor, F. A. Z., Kharroubi, S., . . . Nasreddine, L. (2021). Erosion of the Mediterranean diet among adolescents: evidence from an Eastern Mediterranean Country. Br J Nutr, 125(3), 346-356. https://doi.org/10.1017/s0007114520002731

Naja, F., Hwalla, N., Itani, L., Baalbaki, S., Sibai, A., & Nasreddine, L. (2015). A novel Mediterranean diet index from Lebanon: comparison with Europe. Eur J Nutr, 54(8), 1229-1243. https://doi.org/10.1007/s00394-014-0801-1

Obeid, C. A., Gubbels, J. S., Jaalouk, D., Kremers, S. P. J., & Oenema, A. (2022). Adherence to the Mediterranean diet among adults in Mediterranean countries: a systematic literature review. Eur J Nutr, 1-18. https://doi.org/10.1007/s00394-022-02885-0

Rahi, B., Ajana, S., Tabue-Teguo, M., Dartigues, J. F., Peres, K., & Feart, C. (2018). High adherence to a Mediterranean diet and lower risk of frailty among French older adults community-dwellers: Results from the Three-City-Bordeaux Study. Clin Nutr, 37(4), 1293-1298. https://doi.org/10.1016/j.clnu.2017.05.020

Trichopoulou, A., Corella, D., Martinez-Gonzalez, M. A., Soriguer, F., & Ordovas, J. M. (2006). The Mediterranean Diet and Cardiovascular Epidemiology. Nutrition Reviews, 64(10), S13-S19. https://doi.org/https://doi.org/10.1111/j.1753-4887.2006.tb00258.x

Trichopoulou, A., Costacou, T., Bamia, C., & Trichopoulos, D. (2003). Adherence to a Mediterranean diet and survival in a Greek population. N Engl J Med, 348(26), 2599-2608. https://doi.org/10.1056/NEJMoa025039

Ventriglio, A., Sancassiani, F., Contu, M. P., Latorre, M., Di Slavatore, M., Fornaro, M., & Bhugra, D. (2020). Mediterranean Diet and its Benefits on Health and Mental Health: A Literature Review. In Clin Pract Epidemiol Ment Health (Vol. 16, pp. 156-164). © 2020 Ventriglio et al. https://doi.org/10.2174/1745017902016010156

Yaghi, N., Yaghi, C., Abifadel, M., Boulos, C., & Feart, C. (2021). Dietary Patterns and Risk Factors of Frailty in Lebanese Older Adults. Nutrients, 13(7). https://doi.org/10.3390/nu13072188

Round 2

Reviewer 2 Report

  • I disagree with the authors that ‘The sample size is enough powered according to the sample size calculations presented in the manuscript’. The presented in manuscript calculation of sample size is based on week assumption. Especially if a simple survey is conducted as in presented situation. At least 380 participants are needed. Moreover applied ‘snowballing technique’ as a non-probability sampling technique do not guaranteed representativeness, so the results could be biased.  The ‘snowballing technique’ should be used where potential participants are hard to find - this is not the case of the presented study.
  • Self-reported weight and height – must be indicated in limitation section as a potential source of biases.
  •  ‘history of bilateral hip replacement’ as an exclusion criteria – I am still not convinced – it is not always related to the ‘ability to be able to walk without aid’. For the future, the exclusion and inclusion criteria should be reconsider.
  • Authors replay ‘The FFQ used in this study was not validated against other dietary assessment methods’ - surprises me (in negative way) – without it we do know nothing about reliability and credibility of this tool!!! – it should be verified.

Author Response

  • I disagree with the authors that ‘The sample size is enough powered according to the sample size calculations presented in the manuscript’. The presented in manuscript calculation of sample size is based on week assumption. Especially if a simple survey is conducted as in presented situation. At least 380 participants are needed. Moreover applied ‘snowballing technique’ as a non-probability sampling technique do not guaranteed representativeness, so the results could be biased.  The ‘snowballing technique’ should be used where potential participants are hard to find - this is not the case of the presented study.

We thank the reviewer for his insight on this matter. We agree that the snowballing technique had a huge impact on the representativeness of our study and therefore, we agree with the suggestion to add “Preliminary study” to our title.

The new title is as follows: “Association between Lebanese Mediterranean Diet and Frailty in Community-Dwelling Lebanese Older Adults- A Preliminary Study”

  • Self-reported weight and height – must be indicated in limitation section as a potential source of biases.

Following this suggestion, this has been added as a limitation to the study: “Furthermore, measurement and recall errors, and social desirability bias cannot be ruled out as several measures such as dietary data, height and weight were self-reported.”

  • ‘history of bilateral hip replacement’ as an exclusion criteria – I am still not convinced – it is not always related to the ‘ability to be able to walk without aid’. For the future, the exclusion and inclusion criteria should be reconsider.

We thank the reviewer for this comment, and we will consider the inclusion and exclusion in future studies

  • Authors replay ‘The FFQ used in this study was not validated against other dietary assessment methods’ - surprises me (in negative way) – without it we do know nothing about reliability and credibility of this tool!!! – it should be verified.

Although the current FFQ was not validated against other dietary assessment methods, it was designed in a culturally sensitive method where the items were chosen as a comprehensive representation of the Lebanese diet (Pellet and Shadarevian, 1970). This FFQ has been previously and extensively used in different studies to assess dietary patterns among Lebanese population and their relationship with different health outcomes (Jomaa et al., 2016; Matta et al., 2016; Naja, Hwalla, Itani, Baalbaki, et al., 2015; Naja, Hwalla, Itani, Karam, et al., 2015; Naja et al., 2021; Naja et al., 2019; Naja et al., 2020; Naja et al., 2011; Naja et al., 2017). All these studies have shown coherent and consistent results showing that it is a reliable tool. Furthermore, a similar approach approach was used in other studies (Naja et al., 2021; Naja et al., 2020; Naja et al., 2016; Nasreddine et al., 2020; Nasreddine et al., 2006; Nasreddine et al., 2018). Despite this limitation, the FFQ remains one of the most suitable dietary assessment tools, providing information on the participants’ habitual diet over longer periods of time (Caan et al., 1999). Furthermore, as stated previously, the FFQ was administered by trained nutritionists rather than being self-administered. This approach provides several advantages, as self-administration of the FFQ requires a literate population and may result in inconsistent interpretations of the food list and lower response and completion rates, each of which may jeopardize the validity of the data. This important to mention as in our sample, 68.7% had middle school level education and 18% only had university education, indicating a low literacy level.

A limitation has been added to the discussion to answer the concerns of the reviewer.

“Moreover, although the FFQ was not validated in the current population, it has been extensively used in different studies across Lebanon. Despite these limitations of the FFQ method, it remains one of the most suitable dietary assessment tools. In addition, the FFQ was administered by trained dietitians rather than being self-administered. This approach provides several advantages, as FFQ self-administration requires a literate population and may result in inconsistent interpretations and lower response and completion rates, each of which may jeopardize the validity of the data. This is noteworthy to mention as our sample had a low level of education with 68.7% having middle school education.”

Caan, B. J., Lanza, E., Schatzkin, A., Coates, A. O., Brewer, B. K., Slattery, M. L., . . . Bloch, A. (1999). Does nutritionist review of a self-administered food frequency questionnaire improve data quality? Public Health Nutr, 2(4), 565-569. https://doi.org/10.1017/s1368980099000750

Jomaa, L., Hwalla, N., Itani, L., Chamieh, M. C., Mehio-Sibai, A., & Naja, F. (2016). A Lebanese dietary pattern promotes better diet quality among older adults: findings from a national cross-sectional study. BMC Geriatr, 16, 85. https://doi.org/10.1186/s12877-016-0258-6

Matta, J., Nasreddine, L., Jomaa, L., Hwalla, N., Mehio Sibai, A., Czernichow, S., . . . Naja, F. (2016). Metabolically Healthy Overweight and Obesity Is Associated with Higher Adherence to a Traditional Dietary Pattern: A Cross-Sectional Study among Adults in Lebanon. Nutrients, 8(7). https://doi.org/10.3390/nu8070432

Naja, F., Hwalla, N., Itani, L., Baalbaki, S., Sibai, A., & Nasreddine, L. (2015). A novel Mediterranean diet index from Lebanon: comparison with Europe. Eur J Nutr, 54(8), 1229-1243. https://doi.org/10.1007/s00394-014-0801-1

Naja, F., Hwalla, N., Itani, L., Karam, S., Sibai, A. M., & Nasreddine, L. (2015). A Western dietary pattern is associated with overweight and obesity in a national sample of Lebanese adolescents (13-19 years): a cross-sectional study. Br J Nutr, 114(11), 1909-1919. https://doi.org/10.1017/s0007114515003657

Naja, F., Itani, L., Hammoudeh, S., Manzoor, S., Abbas, N., Radwan, H., & Saber-Ayad, M. (2021). Dietary Patterns and Their Associations With the FTO and FGF21 Gene Variants Among Emirati Adults. Front Nutr, 8, 668901. https://doi.org/10.3389/fnut.2021.668901

Naja, F., Itani, L., Hwalla, N., Sibai, A. M., & Kharroubi, S. A. (2019). Identification of dietary patterns associated with elevated blood pressure among Lebanese men: A comparison of principal component analysis with reduced rank regression and partial least square methods. PLoS One, 14(8), e0220942. https://doi.org/10.1371/journal.pone.0220942

Naja, F., Itani, L., Nasrallah, M. P., Chami, H., Tamim, H., & Nasreddine, L. (2020). A healthy lifestyle pattern is associated with a metabolically healthy phenotype in overweight and obese adults: a cross-sectional study. Eur J Nutr, 59(5), 2145-2158. https://doi.org/10.1007/s00394-019-02063-9

Naja, F., Nasreddine, L., Al Thani, A. A., Yunis, K., Clinton, M., Nassar, A., . . . Hwalla, N. (2016). Study protocol: Mother and Infant Nutritional Assessment (MINA) cohort study in Qatar and Lebanon. BMC Pregnancy Childbirth, 16, 98. https://doi.org/10.1186/s12884-016-0864-5

Naja, F., Nasreddine, L., Itani, L., Chamieh, M. C., Adra, N., Sibai, A. M., & Hwalla, N. (2011). Dietary patterns and their association with obesity and sociodemographic factors in a national sample of Lebanese adults. Public Health Nutr, 14(9), 1570-1578. https://doi.org/10.1017/s136898001100070x

Naja, F., Shivappa, N., Nasreddine, L., Kharroubi, S., Itani, L., Hwalla, N., . . . Hebert, J. R. (2017). Role of inflammation in the association between the western dietary pattern and metabolic syndrome among Lebanese adults. Int J Food Sci Nutr, 68(8), 997-1004. https://doi.org/10.1080/09637486.2017.1312297

Nasreddine, L., Ayoub, J., Abbas, N., Abdul Malik, M., & Naja, F. (2020). Postpartum Weight Retention and Its Determinants in Lebanon and Qatar: Results of the Mother and Infant Nutrition Assessment (MINA) Cohort. Int J Environ Res Public Health, 17(21). https://doi.org/10.3390/ijerph17217851

Nasreddine, L., Hwalla, N., Sibai, A., Hamzé, M., & Parent-Massin, D. (2006). Food consumption patterns in an adult urban population in Beirut, Lebanon. Public Health Nutr, 9(2), 194-203. https://doi.org/10.1079/phn2005855

Nasreddine, L., Tamim, H., Itani, L., Nasrallah, M. P., Isma'eel, H., Nakhoul, N. F., . . . Naja, F. (2018). A minimally processed dietary pattern is associated with lower odds of metabolic syndrome among Lebanese adults. Public Health Nutr, 21(1), 160-171. https://doi.org/10.1017/s1368980017002130

 Pellet, P.L. and Shadarevian, S. (1970) Food Composition Tables for Use in the Middle East. American University of Beirut, Beirut, 117.

Reviewer 3 Report

The authors have improved the manuscript, as suggested.

Author Response

The authors have improved the manuscript, as suggested.

We would like to thank the reviewer for the comments that allowed us to enhance our manuscript.